# Supramolecular Adhesive Materials with Antimicrobial Activity for Emerging Biomedical Applications

**DOI:** 10.3390/pharmaceutics14081616

**Published:** 2022-08-02

**Authors:** Changshun Hou, Yung-Fu Chang, Xi Yao

**Affiliations:** 1Department of Biomedical Sciences, City University of Hong Kong, Hong Kong SAR 999077, China; changshou2-c@my.cityu.edu.hk; 2Department of Population Medicine and Diagnostic Sciences, College of Veterinary Medicine, Cornell University, Ithaca, NY 14850, USA

**Keywords:** adhesive materials, supramolecular interactions, wet adhesion, antimicrobial activity, emerging biomedical applications

## Abstract

Traditional adhesives or glues such as cyanoacrylates, fibrin glue, polyethylene glycol, and their derivatives have been widely used in biomedical fields. However, they still suffer from numerous limitations, including the mechanical mismatch with biological tissues, weak adhesion on wet surfaces, biological incompatibility, and incapability of integrating desired multifunction. In addition to adaptive mechanical and adhesion properties, adhesive biomaterials should be able to integrate multiple functions such as stimuli-responsiveness, control-releasing of small or macromolecular therapeutic molecules, hosting of various cells, and programmable degradation to fulfill the requirements in the specific biological systems. Therefore, rational molecular engineering and structural designs are required to facilitate the development of functional adhesive materials. This review summarizes and analyzes the current supramolecular design strategies of representative adhesive materials, serving as a general guide for researchers seeking to develop novel adhesive materials for biomedical applications.

## 1. Introduction

Adhesive materials have emerged as potential interfacial materials to bond different substrates, which can hold materials together in a functional way. Depending on the processing method and formulation, adhesive materials can usually be categorized into glue-type or tape-type [1]. Glue-type adhesives typically require a long curing process from hours to days to establish strong cohesion and interfacial adhesion. This long curing time may limit their applications in time-sensitive scenarios that require rapid and strong adhesion. Tape-type adhesives can be applied instantly and reversibly, but their adhesion strength is relatively low. These adhesive materials are significant in many fields such as electronics, soft actuators, healthcare, and environmental remediation [2]. When adhesive materials are used in biomedical fields, they can be tailored with desirable functions such as tissue repairing and hemostasis by rational designs. Although traditional adhesives or glues used in the biomedical field such as cyanoacrylates, fibrin glue, polyethylene glycol, and their derivatives are effective, they still suffer from some limitations, including the mechanical mismatch with biological tissues, weak adhesion on wet surfaces, biological incompatibility, and lack of functions [3].

Taking advantage of the dynamic and reversible interactions such as hydrogen bonding, *pi-pi* stacking, electrostatic interactions, host-guest interactions, hydrophobic effects, and *van der Waals* interactions [4,5,6], a variety of functional supramolecular adhesive materials have been developed to realize tough bonding on tissues or organs. In contrast to traditional adhesives, these adhesive materials show different physical forms such as glue, tape, and patch, and versatile delivery approaches such as spray, paste, and injection (Figure 1) [7], providing versatilities and convenience for their applications. Currently, non-covalent interactions are widely exploited and proved effective in developing supramolecular polymers and polymer composites with desired mechanical strength, interfacial adhesion, and intelligent, responsive properties. These supramolecular adhesive materials are promising for a range of biomedical applications [8].

Adhesive materials with tailored properties for specific scenarios are necessary since the applied environment is different. For most biological applications, disinfection abilities are crucial for the adhesive materials because they can act as microbial barriers to protect the hosts from infection upon implantation. The complicated biological systems further require the integration of multifunction with the adhesive materials. For example, adhesive materials with suitable tissue adhesion, antimicrobial ability, and conductivity have applications in hemostasis, tissue repair, and bioelectronics. Yet, more efforts are still needed to develop adhesive materials with on-demand function integration to fulfill diverse requirements in biomedical applications.

This review summarizes recent progress on adhesive materials that have been applied in various biological systems. We primarily concentrate on the molecular or structural designs as well as the function development of the adhesive materials to illuminate how they realize the reliable properties in specific tissues or organs. The basic features of the adhesive materials in the biomedical field will be first introduced. We will then focus on the representative supramolecular adhesive materials as non-covalent interactions can be well controlled to realize the desirable properties. Adhesive materials integrated with antimicrobial activity are elaborated for their ability to resist pathogen invasion, which frequently occurs in the biological systems during or post surgeries. Afterward, the emerging biomedical applications of supramolecular adhesive materials are summarized. The challenges and prospects of supramolecular adhesive materials are finally discussed.

## 2. The Features of Adhesive Materials in the Biomedical Field

Adhesive materials have provided great value as alternatives or adjuncts for sutures, clips, staples, and other commercial bio-glues. In recent years, many sophisticated adhesive materials have employed versatile interfacial interactions such as chemical anchors and non-covalent interactions to bind with tissues or organs for the diagnosis and treatment of diseases (Table 1) [7,9,10,11]. These noninvasive strategies promote the development of on-demand adhesion properties in biological systems, making adhesive materials customized for a range of scenarios.

Bioactive polymers with various functions such as hemostasis and sealing are often used to prepare the adhesives for applications in biological systems. Although most adhesives derived from natural polymers have low immunogenicity, which may not induce adverse reactions during the use period, some derivatives from the biopolymers, such as the quaternized chitosan and biogenic collagen, still present poor biocompatibility. Because tissues are mainly composed of water-containing liquid and various proteins such as the glycoproteins in cell membranes and collagens, they will carry abundant functional hydrophilic groups originating from the amino acids [12]. The use of chitosan can improve the wet adhesion ability with the tissue surfaces due to the hydrophilic interactions. In addition, the amino groups of chitosan can be protonated into -NH^3+^ for antimicrobial activity, as well as the recruitment of the red blood cells to assist the formation of clots for blood coagulation, providing a suitable hemostatic effect. Introducing more quaternary ammonium groups on the backbone of chitosan can increase the number of cationic centers, but a large number of quaternization will cause increased toxicity to surrounding cells and tissues. Collagen is another welcomed natural polymer for preparing biofunctional adhesives, as it can activate the platelet adhesion and the following expression of coagulation factors VIII, IX, XI, and XII [13]. As a hydrolysis product from collagen, gelatin can also retain the main biofunctions for the adhesives.

Compared with natural polymers, synthetic polymers can be endowed with advantages that may be weaknesses in natural polymers. For example, surface adhesion, cell behaviors, and biofunctions are all controllable by rational molecular design and material engineering methods. Polyethylene glycol (PEG) is a widely used synthetic polymer for adhesives since it provides intrinsic adhesiveness to tissues and has suitable physiological compatibility. Furthermore, its molecular structure can be engineered with additional groups (e.g., dopamine, aldehyde group) to promote adhesion properties or transformed into hyperbranched morphologies for more available functional groups [14]. For example, ureidopyrimidinone (UPy)-functionalized PEG hydrogel has shown fast, precise, and local drug-releasing and self-healing abilities. The non-covalent interactions between the UPy motifs made the hydrogel injectable ascribed to the shear-thinning behaviors. The combined pH-responsive drug release, self-healing, and injectable properties enabled the PEG-based hydrogel as the minimally invasive material to induce the generation of growth factors for treating myocardial infarction [15]. However, the realization of all the demanded properties, either by natural polymers or synthetic polymers, is tough due to the conflicts in the molecular or structural designs. It is essential to orchestrate the properties that are required in different biomedical applications, which raises huge demand for the combination of chemistry, material, and biological expertise.

Based on the abundant non-covalent interactions and topological connections of the polymer network, adhesive materials prepared by supramolecular interactions show a series of advantages over traditional commercial and covalently crosslinked adhesives, including tunable mechanical properties, tough and reversible adhesion performance, rapid self-healing ability, suitable cargo-loading capacity, and different functions. These properties can be grouped or tuned through unique supramolecular designs to satisfy the application requirements in diverse biological systems. Monofunctional or multifunctional adhesive materials can be obtained accordingly, and one or multiple properties could be focused in response to the complicated requirements of biological systems. In the following, various adhesive materials’ designs will be analyzed to interpret the potential in emerging biomedical applications.

## 3. Supramolecular Adhesive Materials with Antimicrobial Activity

Pathogen transmissions occur everywhere, from public facilities and medical settings to the human body and biomedical implants [16]. Once the pathogens are colonized onto the implants and medical devices in the nosocomial environment, severe safety problems such as infections and other complications would occur to the patients. Once biofilms are formed, much more effort is required because biofilms are much more difficult to destroy than planktonic bacteria, which are not protected by the extracellular polymeric substances (EPS) [17]. Therefore, adhesive materials with a prominent antimicrobial function will extend their application scope. The antimicrobial capability in the adhesives is usually originated from the endogenous or exogenous antimicrobial components, which can help prevent the transmission of pathogens such as bacteria, fungi, and viruses. The antimicrobial adhesives applied in the biological systems also provide infection prevention, which is an important protection for the host during surgical operations. These adhesive materials can be divided into three categories according to their action mechanisms in killing pathogens: biocide-releasing, contact-killing, and stimuli-responsive killing.

### 3.1. Design Adhesive Materials for Release-Killing

Supramolecular polymers, particularly the supramolecular gels, show high loading ratios for a wide range of biocides, such as antibiotics [18,19,20], metal nanoparticles [21,22,23], nitrogen oxide [24,25,26], and essential oils [27,28,29,30]. Due to the weak interactions between the polymeric chains and antimicrobial agents, or the encapsulation of antimicrobial agents in the porous structures, the incorporated antimicrobial agents can be released from the supramolecular polymers into the aqueous medium in a well-controlled way to kill planktonic bacteria and biofilms [31,32]. Generally, the involved action mechanism is that antimicrobial agents interfere with the metabolism of pathogens. They can damage the cell membranes, suppress protein activity, and inhibit nucleic acid replication [32]. Metal nanoparticles such as silver nanoparticles are often incorporated into the supramolecular polymers as a wound dressing due to the broad-spectrum antimicrobial activity and low bio-toxicity to the human body [33]. For example, one type of antimicrobial adhesive, Gel-TA, made from oxidized tannic acid and gelatin, was used to crosslink with the silver nitrate (Figure 2a) [34]. The Gel-TA adhesives showed excellent cytocompatibility and pronounced wet adhesion properties on the tissues through mussel-inspired adhesion chemistry and outstanding antimicrobial activities to bacteria and fungi, resulting from the inherent antimicrobial capacity of tannic acid and silver nanoparticles. However, the used antimicrobial metal nanoparticles in a high dose would present some toxicity to the human body [35]. Antibiotics such as gentamicin, tobramycin, ampicillin, vancomycin, and minocycline are commonly loaded into the adhesives. For example, Hu et al. reported an injectable hydrogel co-loaded with the antibiotic amikacin and anti-inflammatory naproxen (Figure 2b) [36]. This hydrogel showed a couple of combinational properties, including stimuli-responsive releasing behaviors, mechanical integrity, suitable biocompatibility, antimicrobial activity, and anti-inflammation response.

Although antibiotics still play an important role in controlling pathogens’ proliferation, abusing antibiotics could accelerate the resistance to drugs [31,32,37]. To avoid the resistance caused by the abuse of antibiotics, environmentally friendly antimicrobial strategies have been developed. Essential oils derived from plants have been considered green antimicrobial agents due to their suitable biocompatibility, low cost, and broad-spectrum antimicrobial ability [28,38]. Our group recently developed a carvacrol-regulated mucus-inspired adhesive through crosslinking intermolecular hydrogen-bonded polyurea and carvacrol (Figure 2c) [27]. The obtained supramolecular adhesives exhibited suitable transparency, tunable mechanical properties, reusable adhesion behaviors, controlled release, and long-term antimicrobial activity. The content of incorporated carvacrol will cause many variations such as mechanical properties and release behaviors due to the various intensities of hydrogen bonding interactions between supramolecular polymers and essential oils. The adhesive might help address the issues of virus/pathogen-contained aerosols. We then reported another mucus-inspired supramolecular organogel adhesive coating, which could effectively control the pathogen spread by respiratory microdroplets (Figure 2d) [28]. In this supramolecular system, a polyurea scaffold could provide mechanical strength and accommodate different kinds of lubricant oils. The carvacrol could regulate the hydrogen-bonded network and provide antimicrobial functions, and the silicone oil endowed the organogels with self-healable and lubricant properties. The supramolecular organogel coating could adhere to the substrates and provide a wrapping-layer-assisted disinfection mechanism through the released carvacrol. The pathogen-containing microdroplets were spatially wrapped by the lubricant oils, which would increase the release rate of carvacrol for rapid disinfection. Although the biocide-releasing strategy has drawn wide attention and has been frequently used in antimicrobial adhesives, the limited content of antimicrobial agents could hinder the sustained activity of the adhesives, which may require frequent replacement of the adhesives for continuous disinfection [32].

### 3.2. Design Adhesive Materials for Contact-Killing

Adhesive materials with contact-killing properties can circumvent the limited content of incorporated antimicrobial agents. They realize the disinfection by contacting the surface of pathogens using their antimicrobial components. The effective antimicrobial moieties usually consist of polycations, antimicrobial peptides, and antimicrobial enzymes, which can covalently or non-covalently link to the polymeric chains [37,39]. The antimicrobial mechanisms of contact-killing are diversified, mainly including the cell membranes’ physical lysing or charge disruption [32]. However, the contact-active adhesives usually suffer because the bacteria within the infected tissues or protected by EPS may not come into contact with the antimicrobial components. Therefore, the adhesive should be able to release its components to penetrate the biofilms for disinfection. Gan et al. developed a tough supramolecular hydrogel adhesive by an interpenetrated polymeric network, which included terpolymers and quaternized chitosan (Figure 3a) [40]. The dual crosslinking (covalent and non-covalent) endowed the hydrogel with strong and tough mechanical properties. The monomer of methacrylamide dopamine enabled the hydrogel to contact and adhere to the fibroblasts. Moreover, the quaternized chitosan provided intrinsic antimicrobial abilities for supramolecular adhesives. Therefore, this type of contact-active supramolecular hydrogel could be used for tissue regeneration and wound disinfection. Although these cationic polymers exhibited effective antimicrobial activities, the toxicity to mammalian cells is still a concern that could not be ignored. Antimicrobial peptides (AMPs) are composed of cationic and amphiphilic molecules, which are considered an alternative with available antimicrobial activity and cytocompatibility [31,37,41,42]. A recently reported biomimetic surface engineering strategy integrated the mussel-inspired adhesive peptide with bio-orthogonal click chemistry (Figure 3b) [43]. The dibenzylcyclooctyne (DBCO)-modified AMPs and DBCO-modified nitric oxide generating species of 1,4,7,10-tetraazacyclododecane-1,4,7,10-tetraacetic acid (DOTA) chelated copper ions are clicked onto the surfaces for durable antimicrobial properties and ability in long-term resistance of adhesion/activation of platelets. The engineered surface can therefore be applied to many biomedical devices to prevent infection and thrombosis. In addition, it is reported that the antimicrobial capabilities of the peptide-based supramolecular hydrogels are primarily influenced by the side chain length, charge density, and amphiphilicity of the AMPs [31].

### 3.3. Design Adhesive Materials for Stimuli-Responsive Killing

External physical or chemical changes can activate stimuli-responsive antimicrobial properties of the adhesive materials. This type of adhesive material can usually generate hyperthermia or active molecules such as the reactive oxygen under the trigger by light, electricity, magnetism, and ultrasound [32,37,44]. Some adhesives could also be prepared with responsive abilities to environmental factors such as temperature, pH, and humidity [45,46,47]. The stimuli-responsive ability endows the adhesives with controllable antimicrobial activity that may reduce the development of pathogen resistance. Adhesives integrated with photothermal and photodynamic antimicrobial functions have shown prominent values in numerous biomedical applications. A wide range of nano-agents such as gold nanoparticles, carbon nanomaterials, and metal-sulfide nanomaterials have excellent light-to-heat conversion capacity [48,49,50,51], while the photosensitizers could also lead to the damage of membranes and DNA molecules of the pathogen [52].

Taking advantage of the UPy-assisted supramolecular interactions, we have developed one supramolecular nanocomposite adhesive coating (Figure 4a) [50]. The adhesive coating was mechanically induced by the assembly of powders, which contained the mixture of the siloxane-modified gold nanorods and siloxane-derived crosslinkers. The nanocomposite adhesive coatings exhibited fast preparation, strong adhesion, high stiffness, rapid disinfection ability, and remarkable disinfection effectiveness over 99.9% to different kinds of multi-drug-resistant bacteria within 6 s near-infrared irradiation (NIR). Liang et al. reported an antimicrobial adhesive hydrogel through a dual-dynamic-bond crosslinked mechanism consisting of ferric iron, protocatechualdehyde, and quaternized chitosan (Figure 4b) [51]. The dynamic properties of catechol-Fe coordination and Schiff base bonds endowed the adhesive with self-healing and on-demand dissolution or removal properties. The quaternized chitosan provided the inherent antimicrobial ability, and the compound of catechol-Fe with NIR properties provided the photo-responsive antimicrobial capability. Despite the photo-induced heating having made significant progress, the generated heat may also damage the local cells and tissues [32]. For adhesives with photodynamic antimicrobial function, the incorporated photosensitizers could transform the light energy into chemical energy to induce the generation of active molecules such as hydroxyl radicals, superoxide, or singlet oxygen for pathogen disinfection. For example, Castriciano et al. reported an antimicrobial scaffold with photosensitizer releasing ability. The scaffold was fabricated from the polypropylene fabrics and loaded with the poly (carboxylic acid)-cyclodextrin/anionic porphyrin through the typical host-guest interactions (Figure 4c) [53]. This method showed a sustained and efficient photodynamic antimicrobial activity to Gram-positive S. aureus and Gram-negative P. aeruginosa. Paul et al. also developed a light-activatable nanospray coating with photodynamic therapy of microbes (Figure 4d) [54]. The sustainable antimicrobial coating from adhesive, UV-resistant, and antimicrobial lignin-integrated abilities of photoluminescence and singlet oxygen generation render the coating promising in phototheranostic applications. Moreover, supramolecular polymers containing photosensitizers usually showed better antimicrobial activity to Gram-positive bacteria than Gram-negative bacteria due to the structural differences in the cell membranes [52,55]. These results demonstrated that the stimuli-responsive properties enabled the adhesive materials to have smart properties.

## 4. Design Supramolecular Adhesive Materials for the Specific Biomedical Applications

### 4.1. Supramolecular Adhesive Materials for Wound Repair

Wound healing is a complicated physiological process that requires many factors to work synergistically. It is sequentially and precisely programmed into four phases: hemostasis, inflammation, proliferation, and tissue remodeling or resolution [56]. Each phase must happen at the appointed time at an optimal intensity; otherwise, the wounds will present impaired healing. Acute wounds and chronic wounds also have different injury mechanisms that may need different therapeutic treatments. The full-thickness skin injury without proper treatment will usually cause serious damage to the body that may induce bacterial infection and other complications for irreversible cutaneous necrosis. In the meantime, it is challenging to achieve scarless skin regeneration with all the recovered appendages, such as the hair follicles and sebaceous glands. Therefore, it is urgent to develop advanced healthcare dressings to cure wounds with functional recovery.

Hemostasis is a key and initial step for healing the wounds and tissues that can activate the following steps. Sutures and staples are the common clinical strategies to close wounds and tissues in surgical operations for wound healing [51,57,58,59]. However, these traditional methods generally cause tissue damage and require a long operation time on the injured tissues, thus increasing the risk of infection [51,60]. The secondary damage may occur if these stitched tools are removed from the tissues. Moreover, these methods are easy to fail when repairing large wounds [57].

Supramolecular hydrogel adhesives are attractive for wound hemostats and healing because they can provide robust mechanical support, fast and strong tissue adhesion, excellent hemostatic effects, and functionalities [11,61,62,63,64,65,66,67,68]. Moreover, they could be endowed with smart properties to monitor the healing process in vivo [47]. Hemostatic materials have shown different mechanisms to promote hemostasis, such as the aggregation of platelets and blood clots, activation of platelets, and coagulation of blood. These mechanisms further improve the hemostatic ability of the adhesives when the hemostatic components are well incorporated. Meanwhile, the strong adhesion ability prevents blood leak from burst pressure. For example, Hong et al. developed a strong hemostatic adhesive hydrogel prepared from the light-induced imine-crosslinked networks, which could be quickly gelled and firmly adhered to the tissues within seconds of UV light exposure (Figure 5a) [69]. The obtained adhesives prevented high-pressure bleeding from the big incision wound of pig carotid arteries and hearts, thus demonstrating high mechanical strength, strong adhesion capacity, and rapid hemostatic effects. Gao et al. also reported one hydrogel-mesh composite with a macroscopic entanglement network, consisting of an interpenetrated PNIPAAm/chitosan crosslinked network and a PET surgical mesh (Figure 5b) [59]. The hydrogel-mesh composite could provide robust interfacial adhesion with tissues via synergetic non-covalent and covalent interactions and maintain stable wet adhesion properties due to the hydrophobic feature of PNIPAAm at the body temperature. The hydrogel-mesh composite demonstrated excellent surgical application potentials in wound hemostasis and healing on the carotid artery, lung, and liver. A variety of protein-based adhesives for wet tissue adhesion and wound closure applications have been recently reported, which were prepared through the multiple supramolecular interactions between cationic supercharged polypeptides and anionic aromatic surfactants (Figure 5c) [70]. The polypeptides adhesives exhibited biocompatibility, biodegradability, and strong adhesion on the soft tissues and stiff substrates. The maximal adhesion strength on hard surfaces was 16.5 ± 2.2 MPa, and the adhesion energy on porcine heart tissues was 260 ± 110 J/m^2^, which was higher than those of most reported protein-based adhesives and covalently crosslinked adhesives. In this case, the strong adhesion and cohesive properties took important roles in sealing the hemorrhagic wounds, resulting from the multiple non-covalent interactions, such as electrostatic interactions, hydrogen bonding, hydrophobic interactions, cation-*π* interactions, and *van der Waals* interactions.

Wound conditions should be taken good care of to promote wound healing because the moisture, airiness, and sterility at the wound site can greatly influence the healing effects [71]. The adhesive closely contacts the wound for moisture absorption, air permeability, and disinfection, leading to sustained wound treatment. An adhesive with long-term adhesiveness, high toughness, and antibacterial ability is fascinating. For example, Ag-Lignin nanoparticles (NPs) as the antibacterial components have been incorporated into the polyacrylic acid (PAA)-pectin hydrogel through multiple supramolecular interactions (Figure 6a) [72]. Furthermore, the Ag-Lignin NPs could form the dynamic balance for the catechol redox due to the reversible quinone-catechol reaction, resulting in long-term and repeatable adhesiveness. Meanwhile, the epidermal growth factor (EGF) could be loaded into the adhesive hydrogel for accelerated wound closure. H&E staining was performed to demonstrate the successful regeneration with an intact and complete epidermis layer.

However, functional regeneration of the skin appendages is still a challenge as the majority of wound dressings used so far could not achieve scarless healing with complete skin components. Qi et al. demonstrated that complete skin healing without reducing skin functions should cover inflammation inhibition, angiogenesis promotion with the expression of vascular endothelial growth factor (VEGF) and EGF, scar prevention through regulating the expression of TGF-β1 and TGF-β3, and the recruitment of mesenchymal stem cells to the wound site [73]. This report provided a basic biomedical theory for the design of adhesives used in wound healing. To restore tissues with physiological activity, an innovative hydrogel scaffold was recently developed with the ability to activate the adaptive immune response in vivo (Figure 6b) [74]. The scaffold could be degraded slowly and stimulate the migration of the IL-33 type 2 myeloid cells. The incorporated D-chiral amino acids could elicit the generation of antigen-specific immunity. Therefore, this finding showed an immunological therapy method to facilitate cutaneous regeneration. Wound repair is more difficult for complicated conditions such as wounds being infected by bacteria. We have recently reported an antimicrobial supramolecular adhesive through the assembly and aggregation of hydrogen-bonded UPy-polyethylenimine (PEI) particulates (Figure 6c) [75]. The hierarchical hydrogen bonds consisting of strong multivalent hydrogen bonds from the UPy motifs and weak hydrogen bonds from the PEI chains endowed the supramolecular adhesives with exudate-sensitive properties. The UPy motifs provided mechanical robustness to the supramolecular adhesive, and the antimicrobial and adhesion ability came from the hydrophilic PEI molecules. The obtained supramolecular adhesives demonstrated strong adhesion ability on animal skins and sustained release of the antimicrobial particulates for wound disinfection. The particulate-aggregated adhesive showed better healing effects on the bacteria-infected wounds than those of commercial biomedical gauze.

**Figure 6 pharmaceutics-14-01616-f006:**
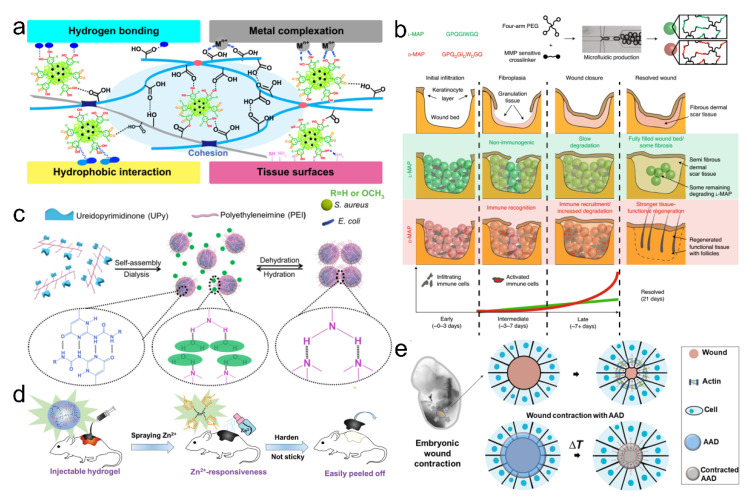
(**a**) The NPs-P-PAA hydrogel could attach to the skin wounds for disinfection and healing. (**b**) Representation of the matrix metalloprotease (MMP) cleavage sequences, amino acid chirality within the crosslinking peptides, and microfluidic formation of the micro-hydrogel that incorporated L- or D-chirality peptides (top row). Compared with L-chirality hydrogel, the D-chirality hydrogel activated the adaptive immune system over time, which caused tissue remodeling and skin regeneration as the adaptive immune system degraded the D-chirality scaffold. Reproduced with permission [74]. Copyright 2021, Springer Nature. (**c**) A supramolecular adhesive through the assembly and aggregation of hydrogen-bonded UPy-PEI particulates demonstrated the strong adhesion ability on animal skins and sustained release of antimicrobial particulates for wound disinfection. Reproduced with permission [75]. Copyright 2020, American Chemical Society. (**d**) Scheme illustration of the wound dressing change facilitated by spraying with zinc ions. Reproduced with permission [76]. Copyright 2020, Royal Society of Chemistry. (**e**) Skin wounds of the chicken embryo (left), in which an actin cable (green) is formed in the cells at the wound edges and contacts the wound. Active wound contraction was enabled by the dressing that adhered to and contracted the wound edges at the skin temperature. Red dashed arrows indicate the contraction.

The new methods of using adhesives to treat wounds were also reported, which were quite different from the conventional drug or cell therapy. These methods have demonstrated improved efficacy in the healing effects. One typical adhesive capable of fast gelation and dissociation in situ has attracted much attention [76]. They synthesized the hyperbranched poly (amino ester) paradigm named HB-PBAE through the Michael addition of dopamine, poly (ethylene glycol) diacrylate, and pentaerythritol triacrylate. The paradigm could be fast gelled upon the addition of Fe^3+^ and removed from the tissues after spraying the Zn^2+^ solution (Figure 6d). The reversible and fast adhesion provided a novel concept for the design of adhesives. Another kind of adhesive called mechanically active adhesive dressing (AAD) has presented the temperature-responsive wound closure [45], inspired by the embryonic wound closure. They designed the adhesive dressings by introducing thermosensitive polymers, which could mechanically contract the wounds upon exposure to the skin temperature (Figure 6e). Following that, Hu et al. further integrated the immune regulative ability into such a dressing for synergetic therapies for wounds [46]. All these results suggested that the adhesives could be integrated with sufficient functionalities, which have promising potential in treating wounds.

### 4.2. Supramolecular Adhesive Materials for Tissue Sealing

Bioadhesive supramolecular hydrogels are often used as tissue adhesives because of their comprehensive advantages, including the noninvasive use method, mechanical properties, and biocompatibility. They have three-dimensional hydrophilic crosslinked networks, which mimic the construction of extracellular matrix and are capable of loading a variety of natural or synthetic materials such as bacteria, exosomes, and drugs [57,77,78,79,80,81]. The adhesives can be customized for injectable, paintable, and attachable use, facilitating the operations on different tissues with irregular wounds during surgical procedures. They have shown enormous potential as alternatives or adjuncts for sutures, clips, staples, and other commercial bio-glues. Tissue adhesives have been well applied in many body parts, whereas the specific requirements vary a lot. In the following, the demonstrations of tissue adhesives are primarily on the common soft tissues for illuminating their values in biomedical applications.

Adhesive materials are welcomed in the field of tissue repair and regeneration, including cartilage repair, bone regeneration, cardiac repair, and nerve repair. The structural similarity and mechanical compliance with soft tissues and strong adhesion properties provide the convenience of operation for the surgeons [77,82,83,84,85,86]. A polydopamine-chondroitin sulfate-polyacrylamide hydrogel was developed as the tissue adhesive (Figure 7a) [87]. Due to the non-covalent interactions and physicochemical properties, the hydrogel adhesives exhibited excellent mechanical toughness and resilience, superior adhesion properties, and suitable cell affinity, which were suitable for chondrocyte growth and cartilage regeneration even in a growth-factor-free condition. The in situ formation of bioadhesives at the target sites in the dynamic biological system usually requires a prolonged gelation time, which may result in poor adhesion and considerable loss of functionalities. A series of bioadhesives that could be formed in situ within seconds were recently prepared, which could realize robust adhesion and better tissue repair on tissues [88,89,90]. Bian et al. utilized a thiourea-catechol reaction to construct a redox polymeric system with strong adhesion properties (Figure 7b) [88]. Compared with classical bioadhesive hydrogels, the formed adhesive hydrogel showed enhanced mechanical properties, exceedingly short curing time (within 5 s), and pH-independent gelation behavior, indicating the excellent promotion of gastric ulcer healing. After further introducing thiol-functionalized 4-arm-PEG into the same polymer system, they recently developed another bioadhesive that could be gelled instantly for acute endoscopic hemostasis in the upper gastrointestinal [89]. They also developed a powder-type self-gelling bioadhesive, which was prepared by the electrostatic interactions of the PEI and polyacrylic acid (PAA) [90]. The physically crosslinked adhesive powder could form into a bulk adhesive within 2 s, which effectively sealed the damaged stomach and intestine and treated the gastric perforation (Figure 7c). These achievements demonstrated an instant and strong adhesion that was in high demand for biomedical applications. Moreover, it frequently occurs that the adhesives are misplaced on dynamic tissue surfaces. Repositioning the misplaced adhesives during surgical operations could seriously cause secondary damage to the tissues. A recent tape-type adhesive was developed to solve the issue, which showed reversible adhesion in the initial stage and robust adhesion in the long term (Figure 7d) [1]. They used the electrical oxidation method to control the adhesion on the tissues by regulating the oxidation of catechol into catechol quinone, which realized strong physical interactions with tissues in seconds and gradually formed the covalent interactions in hours. These traits made the tape valuable in tissue sealing applications. This tape is promising that could function in challenging surgical conditions.

Soft bioadhesive hydrogels are similar to the extracellular matrix in terms of their physicochemical properties, which provide an immune-regulatory microenvironment for the survival and growth of cells or other active materials. They are ideal to be applied as vehicles since intimate contact with the tissues can improve the delivery efficiency through the proposed administration routes [91,92,93,94,95,96,97,98,99,100,101,102]. For example, Zhang et al. reported a self-adhesive drug-loaded bandage for nerve repair and regeneration (Figure 8a) [100]. The self-adhesive bandage protected the injured nerve via a side-close wrap and directionally and continuously released the nano-drugs to the specific locations. Moreover, this self-adhesive bandage demonstrated biocompatibility, biodegradability, improved therapeutic efficiency, and decreased side effects in the nerve regeneration process. In general, traditional drug therapy has some disadvantages, such as the limited drug loading capacity, which urges researchers to utilize supramolecular adhesive gels to encapsulate the living cells, bacteria, or their subunits for producing sufficient therapeutic agents. The active components could continuously secrete substances for treatment [103] and have shown abilities to activate cell migration, infiltration, proliferation, and differentiation for improved treatment effectiveness [81,104]. For example, Dong et al. reported an injectable supramolecular hydrogel adhesive as a cell delivery vehicle for cardiac cell therapy, which was prepared from the chitosan-graft-aniline tetramer and dibenzaldehyde-terminated poly (ethylene glycol) at the physiological conditions [105]. The obtained hydrogels showed excellent conductivity, biocompatibility, biodegradability, antimicrobial activities, and adhesion properties on the tissues. The conductive adhesives also showed a controllable release rate of C2C12 myoblasts and H9c2 cardiac cells. These advantages made the injectable conductive adhesives excellent cell delivery carriers for cardiac tissue repair. Through encapsulating the mesenchymal stem cells (MSCs) into an alginate-based hydrogel biomaterial, the engineered cell-laden adhesive with tunable mechanics can be developed for injection and repairing craniofacial bone tissues (Figure 8b) [106]. Koivusalo et al. also reported one tissue adhesive that could be used for corneal regeneration, which was prepared via grafting dopamine moieties onto the hydrazine-crosslinked hyaluronic acid (HA-DOPA) (Figure 8c) [97]. The HA-DOPA hydrogels exhibited excellent mechanical properties and tissue adhesion properties. The encapsulated human adipose-derived stem cells in HA-DOPA showed suitable proliferation and cellular morphology, while the limbal epithelial stem cells expressed typical limbal epithelial progenitor markers to promote corneal regeneration. These features made the type of tissue adhesive an effective cell vehicle for corneal regeneration.

### 4.3. Supramolecular Adhesive Materials for Bioelectronics

Traditional stiff electrodes cannot meet the physicochemical properties of biological tissues, particularly for the mechanical properties and biocompatibility, increasing the risk of tissue damage and inflammation [107]. Due to the tunable mechanical properties, excellent adhesion properties, high conductivities, and biocompatibility, conductive adhesive materials have attracted widespread attention in the field of bioelectronics [108,109,110,111,112,113]. These adhesive materials aim to form conformal and stable adhesion to both the tissues and bioelectronic devices, capable of converting various action/physiological signals into electrical signals without attenuation. The superior adhesion properties make the adhesive materials applicable to monitor human activities in real-time, through which the accurate signals can reflect the physical conditions for diagnostic and therapeutic purposes [112,114,115]. Adhesive materials with conductivity are roughly divided into electrically conductive and ionically conductive adhesives. The electrically conductive adhesives are usually composed of inherent conductive polymers, such as poly (3,4-ethylenedioxythiophene) (PEDOT), polypyrrole (PPy), and polyaniline (PANI), and conductive fillers, including metal nanoparticles, carbon nanotubes, graphene, and liquid metal. Ionically conductive adhesives are usually composed of intrinsic charged polymers, such as polyelectrolytes and poly(ionic liquid)s, and free charged ions, such as ionic liquids and salts in the polymeric networks [109,110,111,112,116,117].

Adhesive materials with conductive properties have broad application prospects, particularly as strain- or pressure-type sensors to reflect health conditions [118,119,120,121,122,123,124,125,126]. The sensitivity and fatigue resistance of adhesive materials are two important indexes for wearable sensors. High-performance adhesion can provide conformal and stable contact with the underlying surfaces for accurate signal output. For example, Wang and coworkers developed a tough and electrically conductive hydrogel adhesive made from an interpenetrating hydrogen bonding crosslinked network comprising polyaniline (PANI) and poly (acrylamide-co-hydroxyethyl methyl acrylate) (P-(AAm-co-HEMA)) (Figure 9a) [127]. This conductive hydrogel showed excellent mechanical strength, toughness, fatigue resistance upon cyclic deformation, high sensitivity, suitable linearity to the strain, and biocompatibility, which could be directly attached to the diverse human parts as strain sensors for accurate and reliable detection of human activities, including wrist bending, pulse beating, and speaking.

Moreover, physiological signals, including electrocardiography (ECG), electromyography (EMG), and electroencephalography (EEG), could also be monitored by the use of supramolecular adhesives [111,112,120,128,129,130,131]. Xue et al. reported a hydrogel adhesive with reversible adhesion for human-electronics interfaces (Figure 9b) [128]. This adhesive demonstrated an excellent interfacial toughness higher than 400 J/m^2^ on the porcine skin, which was ascribed to the rational supramolecular design. The synergy of hydrogen bonding and chemical anchorage contributes to robust interfacial adhesion. The effective energy dissipation from the dissociation and reconstruction of non-covalent bonding resulted in suitable bulk mechanical properties. Due to the conformal attachment to the skin, robust adhesion in dry and wet states, and superior biocompatibility, this hydrogel adhesive could be used as wearable epidermal electronics for EMG measurement. In addition, flexible wearable devices based on multifunctional sensors have been developed to implement human–machine interactions [111,112,114,115,116,117,132,133]. Zhang et al. reported skin-like supramolecular polymeric networks with ionic conductivity and self-healable properties, which were prepared from supramolecular double networks. The double network consisted of polycarboxylic acid and zwitterionic networks, in which the hydrogen bonding and ionic bonding interactions took the major roles [126]. Such supramolecular adhesive demonstrated excellent mechanical robustness, suitable adhesion, and proton-conductive properties. The nonlinear mechanical responsiveness and supramolecular interactions made the adhesive suitable sensing properties in various applications. The supramolecular networks had low mechanical hysteresis and conformal adhesion, suggesting that they could be applied as multifunctional electronic devices (detection of strain, pressure, and temperature) with high sensitivity and durability for monitoring the human body movement and physical conditions (Figure 9c).

When the adhesive materials are applied in the physiological environment, the requirements get higher since multiple functions should be combined in one adhesive system to ensure successful applications. It is vital to realize conformal, stable, and conductive wet interfacial adhesion between bioelectronics and biological tissues because the complicated physiological environment for the realization of reliable signal sensing, including electrical stimulation and recording in bioelectronic devices, was difficult [107,134,135,136]. Adhesive materials are recently reported to be used as bioelectronics–tissue interface material with diagnostic and therapeutic capabilities for a wide spectrum of diseases and disorders. Many available electronics have suitable performance in the external environment but have failed in vivo studies, while the adhesives can serve as the supporting matrix to protect them from being disabled in physiological conditions. These achievements significantly push forward the development of supramolecular adhesives in biomedical fields. The mechanical properties, wet adhesion ability, conductive properties, durability, and biocompatibility of the adhered bioelectronic devices must be well considered for the specific application scenarios. Deng et al. reported an electrical bioadhesive interface comprised of a graphene nanocomposite hydrogel, which was prepared by the two interpenetrated crosslinked hydrogel networks: a GO-PVA hydrogel network and a PAA hydrogel network (Figure 10a) [130]. The bioadhesive introduced supramolecular interactions to improve the mechanical properties. The dense hydrogen bonds were generated within the two interpenetrated networks, which provided excellent mechanical robustness. In the meantime, the bioadhesive interface facilitated the rapid absorption and removal of the interfacial water for strong physical and covalent interactions with underlying wet surfaces, leading to high interfacial adhesion properties. Further converting GO to reduced-GO could improve the electrical conductivity of the adhesive without compromising the mechanical properties. This electrical bioadhesive demonstrated rapid adhesion, and the interfacial toughness on the heart reached over 400 J/m^2^. The system could also record the ECG signals in vivo, verifying that the bioadhesive could be used for bioelectronics. Therefore, the stable bioelectronic interface promoted the intimate contact of biological tissues and bioelectronic devices and thus showed superior functionality under complicated physiological conditions.

Implantable devices adhered by the bioadhesives have been applied in many aspects, which can implement cell induction and differentiation, neurological signals detection, diagnostics, and therapy, through their interactions with the biological tissues [137,138,139,140]. For example, Gan et al. reported a nanocomposite hydrogel adhesive system in which the conductive nanosheets consisted of PEDOT and polydopamine-functionalized graphene oxide (PSGO), and they were incorporated into a polyacrylamide hydrogel (Figure 10b) [141]. The adhesive exhibited excellent and sustained adhesion properties on diverse solid substrates and soft tissues owing to the multiple non-covalent interactions between the catechol units and substrates. Moreover, the excellent mechanical properties, improved conductivity, and suitable biocompatibility made this hydrogel used for electronic skin to monitor action and physiological signals. In addition, this conductive hydrogel electrode could electrically stimulate cell behavior and was also implanted into a rabbit’s head to detect EEG signals in vivo when the rabbit was jumping and chewing. Sasaki et al. also reported a highly conductive and elastic bioelectrical device assembled by a hybrid hydrogel adhesive. The PEDOT/PU composites were firmly bonded to a double-network hydrogel adhesive [142]. The electrical device showed high electrical conductivity and stability under repetitive mechanical stretching and bending, probably due to the superior adhesion on the interfaces. The bioelectronic device showed abilities to promote the proliferation and differentiation of neural and muscle cells, representing an approach for developing electrical stimulation and spatially controlled cell cultures. Apart from the ability for bioelectronics, the bioadhesive interfaces could also have additional functions such as hemostasis and wound sealing, which have been realized by a hydrogel adhesive with a physical and chemical crosslinking mechanism (Figure 10c) [143].

**Figure 10 pharmaceutics-14-01616-f010:**
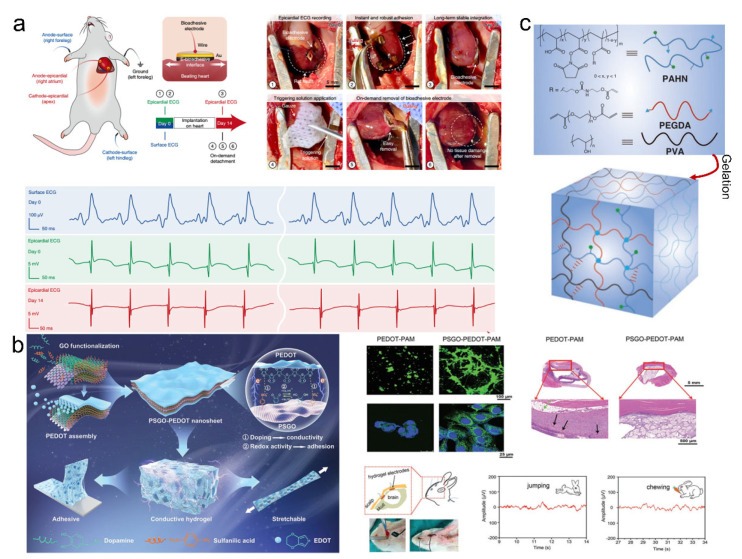
(**a**) The bioadhesive interface with rapid and strong adhesion ability for in vivo recording the ECG signals. The numbers indicated the procedures, including the in vivo epicardial ECG recordings of a rat heart on day 0 and day 14 and on-demand detachment of the bioadhesive electrode on day 14 after implantation. Reproduced with permission [130]. Copyright 2021, Springer Nature. (**b**) A nanocomposite hydrogel electrode could electrically influence the cell behavior and detect the EEG signals. Reproduced with permission [141]. Copyright 2020, John Wiley and Sons. (**c**) The polymer precursor mixture solution is composed of functional PAHN polymer bearing N-hydroxysuccinimide ester and acrylate moieties, PEGDA, and biopolymer, which can be gelatinized by UV irradiation. Reproduced with permission [143]. Copyright 2021, John Wiley and Sons.

## 5. Conclusions

In this review, we elaborate the representative design strategies on the adhesive materials to realize specific functions for emerging biomedical applications such as wound repair, tissue sealing, and bioelectronics. Antimicrobial activity is one of the critical and basic properties of adhesive materials used in biological systems, which can prevent the accommodation of pathogens and facilitate long-term applications. Many emerging adhesive materials for different biomedical applications have been integrated with antimicrobial activity. It is therefore important to understand how to integrate the antimicrobial activity with specific functions in the adhesive materials for better properties. To prepare adhesive materials with antimicrobial activity, the integration of biocompatible polymers with endogenous or exogenous antimicrobial components is widely adopted. From the perspective of biomedical applications, tough and reversible adhesion on wet tissues or organs is usually expected. The specific functions can assist the adhesive materials in fulfilling the specific requirements in the biological systems.

Although the customized adhesive materials for the specific biological requirements are promising, some challenges remain, such as the trade-off between adhesion properties and functions. The complex biological systems may also disable the properties of the adhesive materials. Therefore, the structural or molecular designs should be rationally arranged according to the desired application scenarios. In recent years, supramolecular strategies such as the non-covalent assembly have emerged to develop diversified adhesion and functional properties, which can promote the adhesive materials to target the specific requirements in different tissues or organs. We envision that the adhesive materials integrated with antimicrobial activity can further protect the hosts and thus help the clinical treatment. The development of functional adhesive materials according to the application scenarios could play an important role in future biomedical applications.

## Figures and Tables

**Figure 1 pharmaceutics-14-01616-f001:**
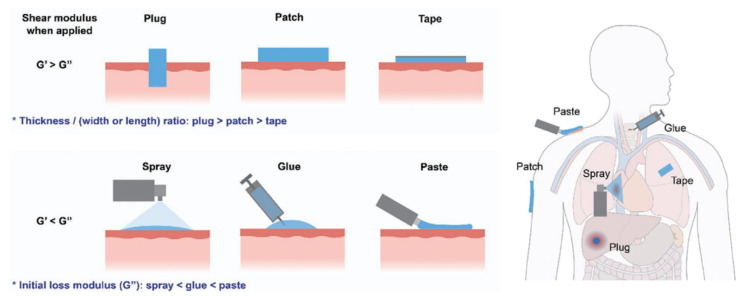
Delivery approaches and physical forms of adhesive materials. Examples show that the various forms of adhesives can be used in different tissues or organs. Reproduced with permission [7]. Copyright 2021, John Wiley and Sons.

**Figure 2 pharmaceutics-14-01616-f002:**
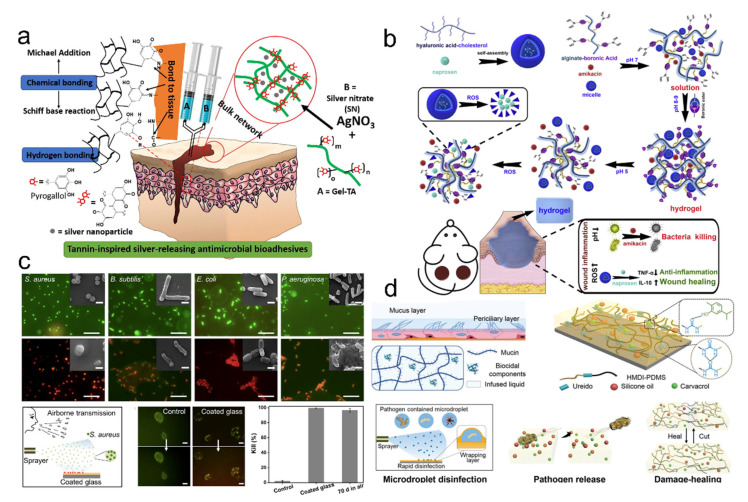
Biocide-releasing of antimicrobial adhesives. (**a**) Adhesive Gel-TA was prepared from the tannic acid and silver nanoparticles, demonstrating wet adhesion properties and inherent antimicrobial activity. Reproduced with permission [34]. Copyright 2018, Elsevier. (**b**) Injectable hydrogels contain the antibiotic amikacin and anti-inflammatory naproxen, which show stimuli-responsive releasing behaviors, mechanical integrity, suitable biocompatibility, antimicrobial activity, and anti-inflammation response. Reproduced with permission [36]. Copyright 2020, Elsevier. (**c**) Carvacrol-regulated supramolecular adhesives with long-term antimicrobial performance. Scale bars of the bacteria staining images and inset SEM images were 10 μm and 1 μm, respectively. Scale bar of microdroplets was 50 μm. Reproduced with permission [27]. Copyright 2020, American Chemical Society. (**d**) Supramolecular organogels incorporated with carvacrol and silicone oil have microdroplet disinfection, adhesion, pathogen-release, and damage-healable properties. Reproduced with permission [28]. Copyright 2021, John Wiley and Sons.

**Figure 3 pharmaceutics-14-01616-f003:**
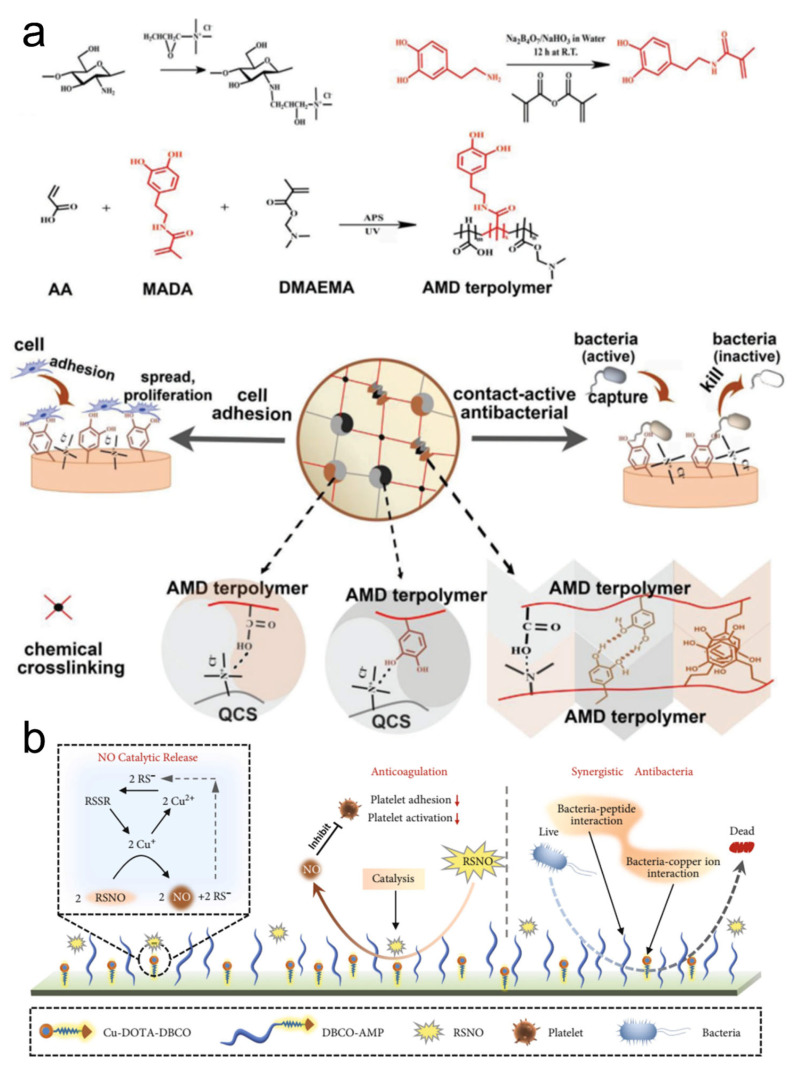
(**a**) A strong and tough hydrogel adhesive could promote the adhesion of fibroblasts, which was derived from the methacrylamide dopamine. The intrinsic antimicrobial properties originated from the quaternized chitosan molecules. Reproduced with permission [40]. Copyright 2019, John Wiley and Sons. (**b**) Realization of anticoagulation and synergistic antibacterial properties of Cu-DOTA&AMP surface.

**Figure 4 pharmaceutics-14-01616-f004:**
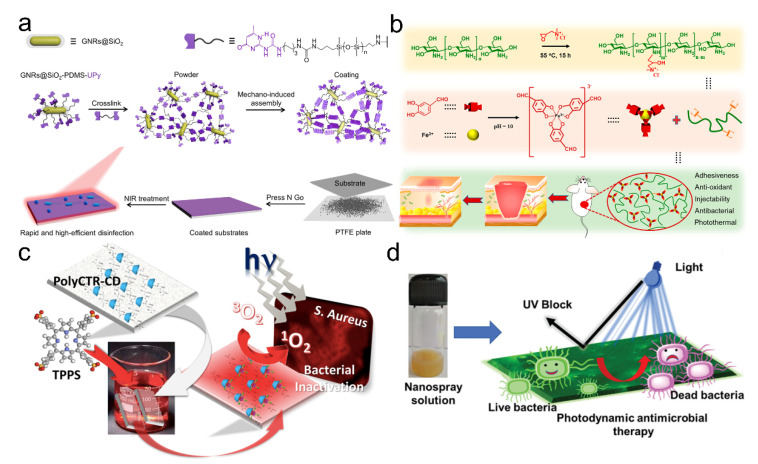
(**a**) A supramolecular nanocomposite adhesive coating was mechano-induced assembled from powders, showing rapid and remarkable photothermal disinfection effectiveness upon NIR irradiation. Reproduced with permission [50]. Copyright 2021, American Chemical Society. (**b**) A dynamically crosslinked antimicrobial adhesive hydrogel in which the quaternized chitosan provided the inherent antimicrobial ability and catechol-Fe with NIR properties contributed to the photo-responsive antimicrobial capability. Reproduced with permission [51]. Copyright 2021, American Chemical Society. (**c**) Antimicrobial polypropylene fabrics assembled by host-guest interactions possess sustained release of photosensitizers and efficient photodynamic antimicrobial activity to Gram-positive S. aureus and Gram-negative P. aeruginosa. Reproduced with permission [53]. Copyright 2017, American Chemical Society. (**d**) Schematic representation of the UV blocking and antimicrobial photodynamic activity using the coating. Reproduced with permission [54]. Copyright 2021, Royal Society of Chemistry.

**Figure 5 pharmaceutics-14-01616-f005:**
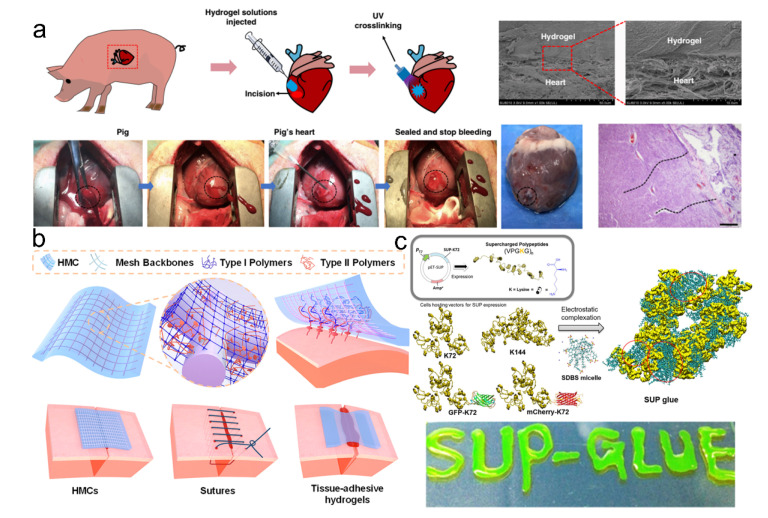
(**a**) A hemostatic adhesive hydrogel showed quick gelation and strong adhesion to the tissues within seconds after UV light exposure, which could prevent high-pressure bleeding from the big incision wound of pig hearts. Scale bar of the tissue staining image was 200 μm. (**b**) The hydrogel-mesh composite with robust adhesion on tissues via non-covalent and covalent interactions, showing excellent wound hemostasis and healing. Reproduced with permission [59]. Copyright 2021, National Academy of Sciences. (**c**) Polypeptides adhesives assembled by multiple supramolecular interactions showed suitable biocompatibility, biodegradability, and strong adhesion on the soft tissues and stiff substrates.

**Figure 7 pharmaceutics-14-01616-f007:**
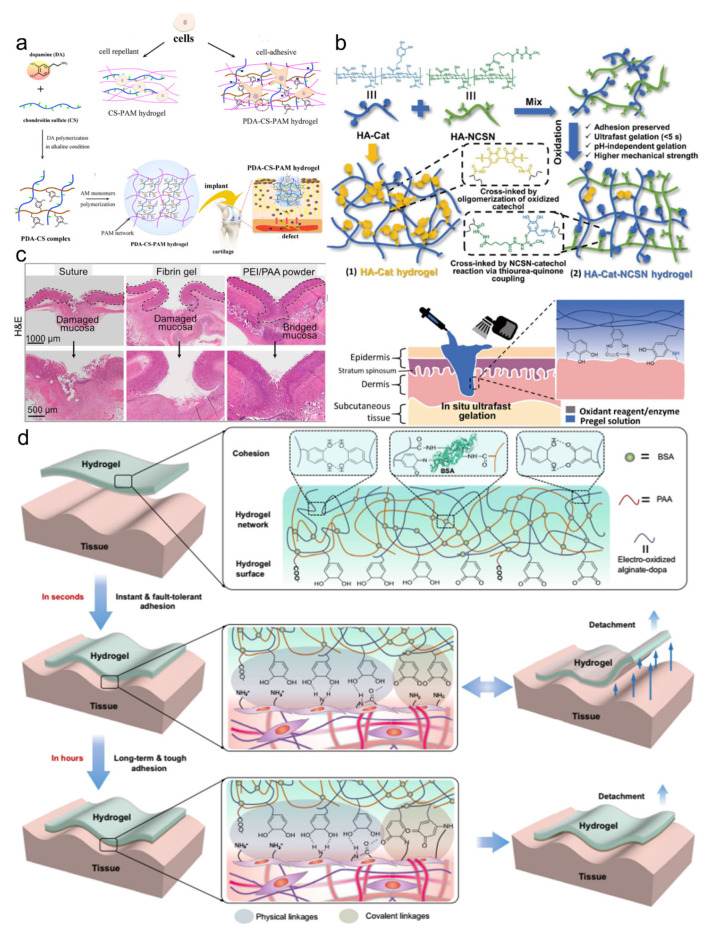
(**a**) Chondrocyte growth and cartilage regeneration were promoted by a growth-factor-free polydopamine-chondroitin sulfate-polyacrylamide hydrogel. Reproduced with permission [87]. Copyright 2018, American Chemical Society. (**b**) Schematic illustration of the fabrication process and mechanisms for the adhesion. Reproduced with permission [88]. Copyright 2020, American Association for the Advancement of Science. (**c**) H&E staining of the gastric wounds treated by different bioadhesives was compared. (**d**) Schematic illustration of the adhesion mechanism on the tissue surfaces.

**Figure 8 pharmaceutics-14-01616-f008:**
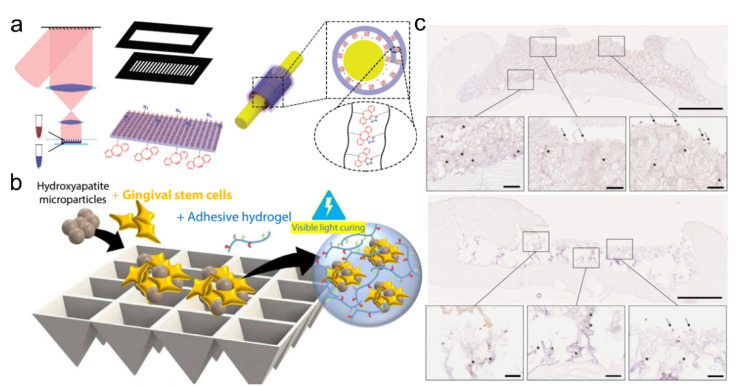
(**a**) A self-adhesive drug-loaded bandage showed the ability of directional and continuous releasing of the nano-drugs to specific locations. (**b**) Schematic of the forced aggregation process to prepare gingival MSCs and hydroxyapatite (HA) aggregates. Reproduced with permission [106]. Copyright 2020, American Association for the Advancement of Science. (**c**) HA-DOPA showed suitable proliferation and cellular morphology, which could serve as an effective cell vehicle for corneal regeneration. The higher magnification images showed encapsulated cells (*) and cells on the surface of the implant (arrows). The circle represents the surface of hydrogel implants. Scale bars in the whole cornea images are 1 mm, and 100 μm in the insets. Reproduced with permission [97]. Copyright 2019, Elsevier.

**Figure 9 pharmaceutics-14-01616-f009:**
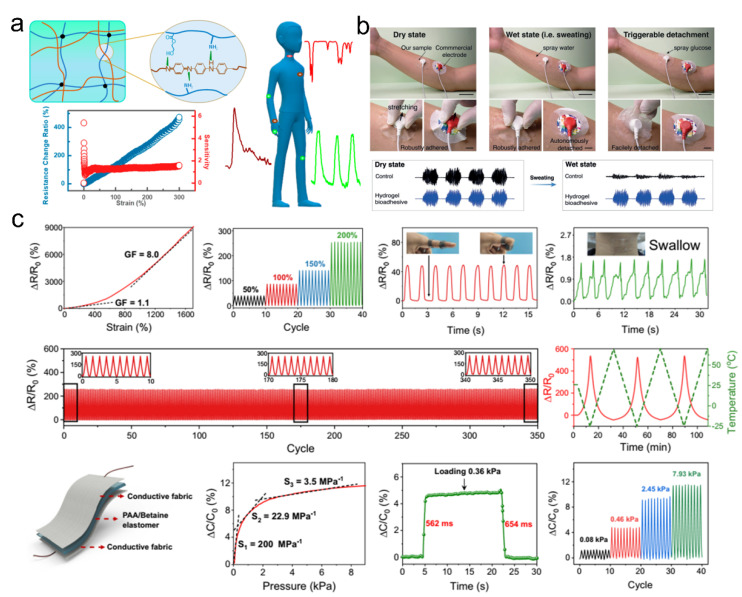
(**a**) The strain sensor for reliable detection of human activities by a tough and electrically conductive hydrogel, which comprised hydrogen-bonding-crosslinked PANI and P-(AAm-co-HEMA). Reproduced with permission [127]. Copyright 2018, American Chemical Society. (**b**) The physiological signal EMG was measured through a hydrogel adhesive with tough and reversible adhesion. Reproduced with permission [128]. Copyright 2021, John Wiley and Sons. (**c**) Multifunctional electronic devices for the detection of strain, pressure, and temperature signals on the human body.

**Table 1 pharmaceutics-14-01616-t001:** Mechanisms of the adhesive materials for bonding the biological tissues or organs.

Adhesion Mechanism on Biological Tissues or Organs					
Type	Adhesive Bond		Topological Connection		
	Chemical Anchor	Non-Covalent Bond	Topological Interlocking	Physical Entanglement	Mechanical Interlock
Pro (s)	High bonding energy with biological tissues or organs; Available under complex physiological environment (e.g., blood, interstitial fluid)	Temporary and reversible bonding with biological tissues or organs; Interfacial interactions (e.g., hydrogen bonding, charge interaction) can be controlled by molecular designs	Adhesion properties (e.g., bonding energy, reversibility) can be controlled by the molecular or structural designs; Stimuli-responsive bonding behaviors (e.g., pH, temperature) can be realized; Available under complex physiological environment (e.g., blood, interstitial fluid)		
Con (s)	Permanent and irreversible bonding with biological tissues or organs	Low bonding energy with wet biological tissues or organs; Easy of debonding under complex physiological environment (e.g., blood, interstitial fluid)	Require complex and time-consuming processing on the surfaces of tissues or organs; Unavailable on smooth and nonporous surfaces of tissues or organs		

## Data Availability

Not applicable.

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
