# Peer review of "Supramolecular Adhesive Materials with Antimicrobial Activity for Emerging Biomedical Applications"

_pharmaceutics, 2022, doi:10.3390/pharmaceutics14081616_

Round 1

Reviewer 1 Report

The manuscript titled “Supramolecular Adhesive Materials with Antimicrobial Activity for Emerging Biomedical Applications” is a well-written review of the biomedical applications of supramolecular adhesive materials. This manuscript provides a good introduction to the design and application of adhesive materials on antimicrobial activity, wound repair, tissue sealing, and bioelectronics. The information in this manuscript is up-to-date and can be helpful for researchers to find the latest works and opinions in this field. This manuscript is overall well-constructed. Following are a few suggestions for the authors to improve the quality of the manuscript. 

  1. The title and conclusion section only focused on the antimicrobial activity of the adhesive materials. However, a substantial part of the manuscript was discussing additional activities such as wound repair, tissue sealing, and bioelectronics. The author may consider modifying the title and conclusion to better reflect the overall topics of the manuscript. 

  1. Some of the figures are blurry, for example, some of the molecular structures in Figure 3, Figure 4, and Figure 7. The author could re-draw the molecular structures to make them clearer for readers. 

Reviewer 2 Report

The manuscript presents a detailed summary of supramalecular adhesive materials regarding their features, antimicrobial activity mechanisms and their specific biomedical applications. The writing is clear. Just some minor revisions for a better flow:

1. Line 279-283, the drawbacks of adhesive with integrated photosensitizers, e.g. 'damage of cell membranes and DNA molecules' should be mentioned after the introduction and advantages of such technique.

2. Line 506-508, the 'however' is confusing. 

3. Line 608, 'they' needs to be more specific. 
